# Nanomaterials as Promising Theranostic Tools in Nanomedicine and Their Applications in Clinical Disease Diagnosis and Treatment

**DOI:** 10.3390/nano11123346

**Published:** 2021-12-10

**Authors:** Wei Zhu, Zhanqi Wei, Chang Han, Xisheng Weng

**Affiliations:** 1Department of Orthopaedics, Peking Union Medical College Hospital, Chinese Academy of Medical Sciences & Peking Union Medical College, Beijing 100730, China; zhuwei9508@163.com (W.Z.); wei-zq15@mails.tsinghua.edu.cn (Z.W.); hanchanglc@gmail.com (C.H.); 2School of Medicine, Tsinghua University, Haidian District, Beijing 100084, China; 3Department of State Key Laboratory of Complex Severe and Rare Diseases, Peking Union Medical College Hospital, Chinese Academy of Medical Science and Peking Union Medical College, Beijing 100730, China

**Keywords:** nanomedicine, nanomaterials, nanoparticles, diagnosis, treatment

## Abstract

In recent decades, with the rapid development of nanotechnology, nanomaterials have been widely used in the medical field, showing great potential due to their unique physical and chemical properties including minimal size and functionalized surface characteristics. Nanomaterials such as metal nanoparticles and polymeric nanoparticles have been extensively studied in the diagnosis and treatment of diseases that seriously threaten human life and health, and are regarded to significantly improve the disadvantages of traditional diagnosis and treatment platforms, such as poor effectiveness, low sensitivity, weak security and low economy. In this review, we report and discuss the development and application of nanomaterials in the diagnosis and treatment of diseases based mainly on published research in the last five years. We first briefly introduce the improvement of several nanomaterials in imaging diagnosis and genomic sequencing. We then focus on the application of nanomaterials in the treatment of diseases, and select three diseases that people are most concerned about and that do the most harm: tumor, COVID-19 and cardiovascular diseases. First, we introduce the characteristics of nanoparticles according to the excellent effect of nanoparticles as delivery carriers of anti-tumor drugs. We then review the application of various nanoparticles in tumor therapy according to the classification of nanoparticles, and emphasize the importance of functionalization of nanomaterials. Second, COVID-19 has been the hottest issue in the health field in the past two years, and nanomaterials have also appeared in the relevant treatment. We enumerate the application of nanomaterials in various stages of viral pathogenesis according to the molecular mechanism of the complete pathway of viral infection, pathogenesis and transmission, and predict the application prospect of nanomaterials in the treatment of COVID-19. Third, aiming at the most important causes of human death, we focus on atherosclerosis, aneurysms and myocardial infarction, three of the most common and most harmful cardiovascular diseases, and prove that nanomaterials could be involved in a variety of therapeutic approaches and significantly improve the therapeutic effect in cardiovascular diseases. Therefore, we believe nanotechnology will become more widely involved in the diagnosis and treatment of diseases in the future, potentially helping to overcome bottlenecks under existing medical methods.

## 1. Introduction

The development of nanotechnology, as well as other relevant methods and materials, has brought about a profound revolution in the field of medicine, giving birth to the new discipline of nanomedicine. With its unique advantages, nanomedicine is playing an increasingly important role in the diagnosis, treatment and prevention of disease [1], due to the rapid development of various nanomaterials. Nanomaterials can be functionalized by binding to specific proteins, reaching specific local regions or releasing drugs in specific environments, which have been a promising diagnostic and theranostic tools in nanomedicine. Diagnostic or therapeutic agents have been designed to aid nanomaterials through biologic barriers to gain access to molecules, regulating molecule interactions and monitoring the changes in the microenvironment [2]. Meanwhile, nanomaterials can be made into different shapes, sizes, compositions, surface chemical characteristics and hollow or solid structures based on their adjustable optical, electronic, magnetic and biological properties [3]. These properties will qualify nanomaterials to have important applications in clinical diseases [4,5].

The application of nanomaterials in tumor diagnosis and treatment is a research focus. Some studies have developed nanoparticles coated with anti-cancer drugs, which can be specifically enriched in tumor areas through the enhanced permeability and retention (EPR) effect of blood vessels, and then degraded in tumor cells after phagocytosis, releasing the wrapped drugs to heal tumors, which can achieve good results [6]. Other studies compound specific proteins or genes on the surface of nanomaterials, which can specifically bind to the target on the surface of tumor cells and kill cancer cells by optical and magnetic means [7]. With SARS-CoV-2 sweeping the world, the lives and health of all humankind are facing a severe threat. At a time when traditional approaches to infection prevention and anti-viral treatment have been overwhelmed by COVID-19, the addition of nanomaterials has played a crucial role in vaccine preparation and virus control for this pandemic. At the same time, many studies have proved that nanomaterials also show unique advantages in the treatment of cardiovascular diseases. This review will focus on functional nanoplatforms in the diagnosis and treatment of diseases.

## 2. Nanoplatforms in the Diagnosis of Diseases

### 2.1. Iron-Based Nanoparticles

Contrast agents are essential in distinguishing normal from abnormal tissues. Iron-based nanoparticles (NPs) have been widely applied in T2-weighted magnetic resonance imaging (MRI) and computed tomography (CT) because of their paramagnetism and X-ray attenuation. Magnetic NPs can affect the T2 relaxation time and cause increased contrast in T2-weighted images. To combine diagnosis with therapy, Hu et al. [8] introduced Prussian blue, hyaluronic acid, NH_2_-PEG and Pt into iron NPs and successfully fabricated a novel nanoagent (PB@FePt–HA-g-PEG NCs). This versatile nanoplatform could not only cause an apparent concentration-dependent darkening effect detected in T2-weighted images, but also effectively ablate 4T1 tumor xenografts with excellent biocompatibility for chemodynamic–photothermal co-therapy. Samira et al. [9] designed magnetic core–shell NPs using a well-shaped o/w emulsion method. Amphiphilic mPEG-PCL co-polymers served as the surface shell for encapsulating the hydrophobic magnetite NPs based on ferric ion. In an MRI diagnostic study, the longitudinal relaxivity value was greatly affected by the aqueous medium’s distance from the magnetite core, which could be extended by a hydrophobic surface coating. This magnetic micelle served as an excellent contrast agent in MRI. However, other studies have used different polymer-coated ferromagnetic NPs in MRI [10,11]. Recently, our team developed an excellent photoacoustic theranostic agent based on CoFeMn dichalcogenide nanosheets [12] (Figure 1). It exhibited the highest photothermal performance with an η of 89.0% for 2D transition metal dichalcogenide materials reported to date. Moreover, 50% of maximum catalytic activity (Michaelis–Menten constant, Km) is attained by CFMS-PVP NSs with 0.26 × 10^−3^ m H_2_O_2_ at 318 K. In in vivo experiments, complete tumor elimination is observed after treatment with CoFeMn dichalcogenide nanosheets at a low dose.

### 2.2. Gold-Based Nanoparticles

Gold shows good ductility and is easily surface modifiable, indicating that it can be crafted into different sizes and shapes. In luminescence and computed tomography, gold nanomaterials are used in imaging due to their strong surface plasmon resonance. However, recently, more studies have developed the composite structure of gold nanoparticles with aptamers and other particles, which can exhibit unique advantages. Based on its coupling plasmonic resonance, a smart core–shell nanoplatform (Au@MnO_2_) was developed and demonstrated to be useful in photoacoustic imaging and photothermal therapy [13]. Gold NPs are good receptors for energy transfer, efficiently quenching fluorescent molecules through Förster resonance energy transfer. As such, they are widely used in the development of “off–on” imaging systems for tumor imaging [14,15]. Yin et al. [16] developed an intelligent “off–on” H_2_O_2_-responsive nanozyme based on gold nanoclusters (AuNCs) loaded into mesoporous silica and manganese dioxide (MnO_2_) wrapped around as a switching shield shell. In acidic tumor microenvironments, this nanomaterial reacted with H_2_O_2_ and generated O_2_, which enhanced photodynamic therapy and brought about excellent MR imaging with a longitudinal relaxation rate of 25.31 mM^−1^s^−1^.

As thermoacoustic signals can be interconverted, the photoacoustic imaging properties of gold nanomaterials have been investigated and used in biomedical imaging [17,18]. Liu et al. [19] prepared glutathione-responsive magnetic gold nanorings through a combination of wet chemical synthesis and layer-by-layer self-assembly to achieve dual-mode tumor localization with nuclear magnetic imaging and photoacoustic imaging, thereby accurately guiding photothermal therapy to tumor cells. For example, an in vitro study reported that the photothermal temperature can reach 80 °C, and subcutaneous tumors can be removed by photothermal therapy. Zhang et al. [20] further realized photoacoustic imaging, CT and photothermal multi-mode tumor imaging using the coupling effect, which was induced by the aggregation of acid-triggered polypeptides and gold NPs. Gold is a new CT contrast agent because of its higher atomic number and stronger X-ray absorption capacity. Liu et al. [21] used CT to study the distribution and tumor uptake of gold nanocrystals in vivo and to track the efficacy of gold nanocrystals in photothermal tumor treatment in mice. Tsvirkun et al. [22] used gold NPs that were covalently linked to cathepsin to target tumors by CT.

Wen et al. [23] designed a new nano-Ag/Au@Au film composite surface-enhanced Raman scattering (SERS) substrate using consecutive layer-on-layer deposition, which rapidly detected creatinine with high sensitivity. Gold NPs were also functionalized on the surface of MoS_2_ self-assembled beta-mercaptoethylamine, a potential biosensor for concanavalin A [24]. Furthermore, the conjugation of gold NPs to specific oligonucleotides served as a nanogenosensor, which detected specific diseases such as trichomoniasis [25] and *Leptospira interrogans* [26].

### 2.3. Carbon Nanoallotrope Nanomaterials

In recent years, nanoscale carbon allotropes have emerged as potential candidates for contrast agents in MRI [27,28] (Figure 2). Wang et al. [29] developed a novel tumor-targeting nanocarrier based on carbon nanotubes. These authors used a Y-shaped peptide either grafted with fluorescein isothiocyanate for fluorescence-based bioimaging or grafted onto partially oxidized carbon nanotubes (OCNTs) using the carboxylic group of the OCNT. Intravenous injection of this nanomaterial was highly efficient in MRI in tumor-bearing mice. Single-walled carbon nanotubes covalently functionalized with carboxylic acid groups were stable in biological media, resulting in a significant decrease in the T2 signal in vitro [30]. When conjugated with an antibody, this nanomaterial exhibited sensitive MRI detection of breast cancer cells by high magnetic resonance r2 relativity [31]. In another study, it was demonstrated that multi-walled carbon nanotubes contaminated with metal (2.57% iron from the growth catalyst) had a significant effect on the T2 transverse relaxation rate of the ^1^H nucleus of water, which could be a useful MRI contrast agent for detecting stem cells [32]. Superparamagnetic iron oxide nanoparticles (SPIONs) could not only be attached to the surface of CNTs [33] but also loaded inside CNTs after synthesis [34].

The discovery of the interesting intrinsic properties of graphene has boosted further research and development for various types of applications, from electronics to biomedicine [35]. Graphene magnetic NP hybrids have attracted much attention due to their excellent biocompatibility and low toxicity. Fe_3_O_4_ NPs can be chemically co-deposited on reduced graphene oxide nanosheets [36], resulting in efficient T2-type graphene-based MRI contrast agents. Carbon nanofiber (GNF) has also been used as an MRI contrast agent. Lu et al. [37] prepared Fe_3_O_4_@GNF@SiO_2_ nanocapsules by synthesizing Fe_3_O_4_ NPs, which were based on carbon nanofiber technology, and used a SiO_2_ coating that required relatively high relaxivity for Fe_3_O_4_@GNF nanocomposites and Fe_3_O_4_@GNF@SiO_2_ nanocapsules, revealing the potential application of these hybrid materials in bioimaging.

### 2.4. Quantum Dot Nanomaterials

Quantum dots (QDs), also known as colloidal semiconductor nanocrystalline, are a semiconductor nanocrystalline with unique photophysical properties. Since their discovery in the 1980s, QDs have moved to the forefront of nanoscience and nanotechnology [38]. QDs have several advantages such as strong photobleaching resistance, wide excitation spectrum, narrow emission spectrum, resistance to chemical degradation and long fluorescence life. In addition, the color of emitted light can change with the particle size, which surpasses the limitations of traditional dyes and realizes the detection of different substances at the same excitation wavelength [39]. QDs can be used to detect various antigenic substances, pathogens and biomolecules, as well as immunomarkers on tissues and cells [40], and have great potential in molecular biology, cell biology, genomics, proteomics, drug screening, biomolecular interactions and other fields [41].

Suzuki et al. [39] developed a QD connected immunoadsorption method using QDs and an antibody, and this detection method could detect IL-6 at a concentration of 0.05 ng/mL. Zhou et al. [42] evaluated the QD-based high-sensitivity troponin detection method, which reduced the detection time from 1 to 2 h to 12 min, and tested the method using whole blood samples from individuals with acute myocardial infarction, and it had a higher diagnostic efficiency and is more suitable for point of care testing (POCT) detection and rapid diagnosis.

### 2.5. Nanopore Sequencing

In the mid-1970s, the emergence of first-generation sequencing, which used the dideoxy chain termination method (Sanger method) [43], initiated the exploration of the human genome. From 1990 to 2003, the Human Genome Project (HGP) rapidly introduced advancements in sequencing and, since then, has developed second- and third-generation sequencing.

Nanopores are nanosized pores formed by the self-assembly of biomolecules or the physical drilling of a solid matrix. They can be classified as biological nanopores, solid nanopores and composite nanopores. Nanopore sequencing is defined as the application that measures DNA or RNA sequence information by recording changes in the current generated by bases passing through the nanopore [44]. As an example of single-molecule sequencing with high throughput, high read length and real-time data output [45], nanopore sequencing is an important advancement that has progressed toward third-generation sequencing. Presently, instruments developed by Oxford Nanopore Technology are beginning to be used in the fields of in vitro diagnostics and public health.

Previously, the competitiveness of nanopore sequencing for next-generation sequencing (NGS) was affected by the relatively high error rate and the need for multiple processing to eliminate sequencing errors [44,46]. However, with the improvement of sequencing accuracy brought about by the maturation and perfection of sequencing, its advantages of high throughput, high read length, real-time data output and direct RNA sequencing have gradually evolved. Orsini et al. [47] developed a method for the rapid detection of the BCR-ABL1 fusion gene based on nanopore sequencing, which is of great significance in the diagnosis of diseases such as acute leukemia. Direct sequencing of the influenza virus genome in the form of primary RNA using nanopore sequencing overtakes the previous process of DNA synthesis by reverse transcriptase and sequencing, thereby avoiding CG bias and preserving the information of base modification in the genome, which is of great value [44]. In addition, nanopore sequencing was also used in the sequencing of the Ebola virus in west Africa [48], and the whole genome sequencing results of the mycobacterium tuberculosis were obtained within 7 to 12.5 h, thereby obtaining information on species identification and drug resistance [49]. As such, nanopore sequencing plays an important role in the sequencing of virus genomes [50] and the further development and improvement of detection techniques that can address virus mutations such as those occurring during the coronavirus disease 2019 (COVID-19) pandemic [51,52].

## 3. Functionalized Nanoplatforms in the Treatment of Disease

### 3.1. Application of Nanoplatforms to Detect Tumors

Despite rapid advances in biomedicine, cancer is one of the leading causes of death, as tumors are difficult to treat. Chemotherapy, radiotherapy and surgery are the main modalities of treatment in clinical settings. However, these modalities have several disadvantages. The main disadvantage of chemotherapy and radiotherapy is the lack of specificity and the onset of severe side effects. Furthermore, an inadequate drug concentration at the tumor site often leads to low anti-tumor efficiency and a high tumor recurrence rate. With the development of nanotechnology, NPs are widely designed for the delivery of anti-tumor drugs in the hope that they can overcome the limitations of traditional therapies.

#### 3.1.1. Characteristics of Nanoparticles

To achieve efficacious tumor treatment, NPs need to have several features. NPs must be biocompatible, bioavailable and stable under physiological conditions to protect functional drug molecules from degradation during delivery and target specific sites without damaging surrounding healthy tissues or cells. These characteristics are largely affected by the physical and chemical properties of NPs [53].

(1)Particle size

The NP size determines the distribution. Betzer et al. [54] reported that NPs as small as the approximate limit size (i.e., E200 nm) could cross the blood–brain barrier through clathrin-mediated endocytosis. Kang et al. [55] reported that smaller polyethylene glycols (PEGs) (≤20 kDa, 12 nm) exhibited significant tumor targeting with minimal to no nonspecific uptakes, while larger PEGs (>20 kDa, 13 nm) accumulated highly in major organs. In addition, the size of NPs usually involves multiple factors. NPs should be small enough to escape the macrophages of the reticuloendothelial system, but they should be large enough to prevent extravasation from normal blood vessels [56]. Meanwhile, for nanoparticles in cancer therapy, sufficient tumor penetration needs a small particle size, while long in vivo circulation time needs a larger particle size [57]. Jasinski et al. [58] found that circulation time increased with increasing RNA nanoparticle size from 5–25 nm, which is the common size range of therapeutic RNA nanoparticles. Sun et al. [57] reported that a nanocluster (RPSPT@SNCs) could preferentially accumulate in tumor tissue and dissociate under extracellular matrix metalloproteinase-2 (MMP-2) to release small micelle formulations (RPSPTs), which possessed favorable tumor penetration and tumor targeting capability to deliver the anti-tumor agent paclitaxel (PTX) into deep regions of solid tumors.

(2)Particle shape

Shape is another important factor when designing NPs for tumor treatment. Zhang et al. [59] reported that rod-shaped gold NPs diffused more rapidly in tumor interstitial fluid than the spherical ones. Arnida et al. [60] showed that PEGylated Au nanorods were distributed within tumors, whereas Au nanospheres and nanodisks were distributed on tumor surfaces. As such, the different shapes of bacteria and viruses seem to allow them to evade the immune response. Therefore, the non-spherical shape that NPs have may contribute to their superior performance over nanospheres of a similar size [56]. Geng et al. [61] demonstrated that filomicelles remained in the circulation for up to one week, which was about ten times longer than their spherical counterparts. In addition, the shape of nanocarriers can regulate the interaction between the cell membrane and NPs. Thus, shape is one of the main factors affecting whether NPs are taken up by the reticuloendothelial system [53,62].

(3)Particle surface characteristics

A previous study has reported that neutral nanocarriers with no net charge have a longer circulation time, which makes them suitable for passive targeting. Negatively charged nanocarriers are removed by Kupffer cells, whereas positively charged nanocarriers are removed by optical ionization [1]. However, tumors, and even different stages of the same tumor, may require nanocarriers with different surface charges. Sriastava et al. [63] reported that NPs containing anionic phosphate groups could target metastatic breast cancer. On the contrary, drugs delivered within sulfonate-functionalized carbon NPs facilitated intracellular transport. In addition, unmodified nanocarriers are usually cleared by macrophages in the reticuloendothelial system because of their hydrophobic surfaces [64]. Therefore, to achieve successful drug delivery, the surface of the nanocarrier should be coated with a hydrophilic material, such as polyethylene glycol (PEG), to minimize opsonization [65].

NPs have several ligands derived from proteins, nucleic acids, peptides and carbohydrates, which are attached to the receptors expressed by tumor cells, and they can mediate the attachment and accumulation of NPs at the tumor site through receptor-mediated endocytosis. For example, carbon nanotubes (CNTs) were coated separately with synthesized drug-conjugated glycoblock co-polymers and folic acid (FA) to obtain an efficient drug delivery platform for dual targeting of glucose transporter protein (GLUT5) and folic acid receptors (FR) in breast cancer [66].

#### 3.1.2. Nanoplatform for Drug Delivery

The unique properties of nanoplatforms overcome the limitations of traditional tumor treatments. Firstly, due to the small size of nanomaterials, NPs overcome biological barriers that traditional drugs cannot penetrate. Another advantage of the small size is that the large specific surface area of the nanomaterial allows more drug molecules to attach [67]. Secondly, NPs can greatly improve the solubility of anti-tumor drugs. Thirdly, the addition of NPs significantly prevents low-molecular-weight conventional drugs from being eliminated by the renal system, thereby increasing drug accumulation in tumors [53]. Another benefit is that NPs prevent the metabolic degradation of drugs during delivery, thereby controlling their toxicity to surrounding tissues.

(1)Liposomes

The implication of liposomes in tumor has garnered significant attention owing to their biocompatibility, safety and high drug loading [68]. Liposomes are composed of phospholipids and cholesterol and other stable components. These components are assembled into vesicles, with a hydrophilic head, hydrophobic tail and cholesterol bilayer membrane [69]. They can carry hydrophilic drugs in their aqueous interior, and hydrophobic drugs can be dissolved in liposomes [70]. The mechanism of liposomes for drug delivery is fusion into the lipid bilayer of cells so that the drugs can be delivered into the cytoplasm. Liposomes effectively protect functional components from environmental degradation, minimize non-specific toxicity of tumor drugs and enhance their accumulation in tumors. In addition, liposomes are retained in the vasculature because the tight junctions of endothelial cells do not allow substances to leak out. However, tumor vessels are leakier, thereby allowing the nanosized liposomes to leak out from the vasculature to the targeted tumor site [71,72]. For example, Doxil, also known as liposomal doxorubicin, the first anti-tumor nanodrug approved by the FDA, uses a PEGylated liposomal formulation. It is approved for breast cancer in the United States and multiple myeloma in Europe and Canada [73].

(2)Polymeric nanoparticles

Polymeric NPs developed as drug delivery vehicles come in many forms, such as dendrimers, micelles and microbubbles. One of the distinctive benefits of polymeric NPs is that they break down into separate monomers that can be easily eliminated from the body through metabolic processes [74].

(a)Dendrimers

As a type of highly branched macromolecule, dendrimers have a low-density interior and a high-density exterior, and they have been widely used in nanomedicine. Compared to traditional polymers, dendrimers are symmetrical monodisperse nanomaterials with an array of functional groups [75]. These properties, combined with the ease of functionalization of the surface groups, allows dendrimers to be developed into drugs. For instance, the loose core of dendrimers can accommodate anti-tumor drugs via non-covalent bonds, whereas a dendrimer–drug complex is assembled by covalently attaching anti-tumor drugs to peripheral functional groups. Thus, several drug molecules can be attached to each dendrimer, and the release of these drug molecules is partly controlled by the type of the functional bond [76]. Several functional bonds are available for the stimulus-responsive release of drug molecules, including esters, hydrazone, carbamates, cis-aconityl bonds and disulfide bonds [77].

(b)Micelles

Micelles are commonly used to carry water-insoluble or poorly water-soluble drugs to tumor cells, as micelles consists of a hydrophobic core and a hydrophilic shell, which allows the loading of hydrophobic micromolecules into the core, while increasing the spatial protection of drugs through the shell [78]. In addition to hydrophobic molecules, hydrophilic drugs and macromolecules, such as nucleic acids, can also be housed within micelles by electrostatic attraction or chemical bonds [79].

The first micelles blocked co-polymer micelles, including hydrophobically assembled amphiphilic micelles, polyion complex (PIC) micelles, and micelles stemming from metal complexation [78]. When these polymers enter the systemic circulation, amphiphilic molecules spontaneously self-assemble into supramolecular core/shell structures, and water-insoluble drugs are pressed into the hydrophobic cores. For example, PEG5K-CA8, developed by Xiao et al. [80], consists of a PEG-cholic acid conjugate that is capable of delivering paclitaxel. The other type of micelle has a hydrophobic core composed of lipids. Musacchio et al. [81] synthesized several stable micelles that utilized strong hydrophobic interactions between double acyl chains. These micelles can solubilize many types of poorly water-soluble drugs.

Micelles have been used to improve the efficacy of anti-tumor drugs. Gener et al. [82] reported that polymeric micelles loaded with zileuton, a tumor stem cell inhibitor, effectively prevented circulating tumor cell (CTC) metastasis in a mouse model (Figure 3). Mutlu-Agardan et al. [83] developed redox-responsive PEG2000-S-S-PTX nanomicelles, which have been used to deliver paclitaxel to chemoresistant breast and ovarian cancer cells under different conditions.

(c)Protein nanoparticles

Protein NPs are biodegradable, non-antigenic and amenable for further functionalization. Wang et al. [84] demonstrated that smart cyanine-grafted gadolinium oxide nanocrystals (Cy-GdNCs) obtained by albumin-based biomineralization were theranostic nanocomposites, with promising properties for trimodal near-infrared fluorescence/photoacoustics/magnetic resonance imaging-guided photothermal tumor ablation. In addition to plant and animal proteins, viral proteins have also been applied to the construction of nanocarriers. Lv et al. [85] reported that multi-functional nanovectors transformed from viral light particles have exhibited excellent properties of active tumor targeting, in vivo tumor imaging and anti-tumor efficacy.

(3)Inorganic nanomaterials

In addition to the aforementioned natural and synthetic polymers and their assembled nanostructures, there are many inorganic nanomaterials that can be used as nanocarriers.

(a)Silicon-based nanoparticles

Silicon-based NPs in solid, hollow, porous or other forms are ideal candidates for the construction of nanoplatforms. Mesoporous silica NPs (MSNPs) have attracted much attention because of their advantages such as large specific surface area, mesoporous structure, controllable pore size, surface functionalization and biocompatibility [86]. To improve tumor-targeted drug delivery, Alizadeh et al. [87] reported that chitosan-coated silica NPs loaded with epigallocatechin gallate (EGCG) improved the cytotoxic effect of EGCG. Gao et al. [88] demonstrated that mifepristone-loaded MSNs coated with the epithelial cell adhesion molecule antibody could specifically target and bind colorectal cancer cells, driving the captured cells into the G0/G1 phase and inhibiting the heteroadhesion between cancer cells and endothelial cells. In addition, Wang et al. [89] demonstrated that the mucoadhesive and drug release properties of MSNPs could be controlled by the number of poly(amidoamine) dendrimers on the nanoparticle surface, holding significant potential for the development of mucoadhesive drug delivery systems for bladder cancer therapy.

(b)Carbon-based nanoparticles

Carbon-based anti-tumor nanoplatforms include a wide range of applications, including fullerene, CNTs and graphene. Due to space constraints, this section only discusses CNTs. As CNTs have a regular structure, strong mechanical strength, metallized properties, large specific surface area, ultra-light weight, high electrical conductivity and thermal conductivity, CNTs are appropriate carriers for drug delivery. CNTs have been demonstrated to remain in lymph nodes longer than spherical nanocarriers [90]. Morais et al. [91] demonstrated that the naringenin-functionalized CNTs showed lower cytotoxicity on non-malignant cells (hFB) than free naringenin, with an improved anti-cancer effect on malignant lung cells (A549) as an in vitro model of lung cancer. Li et al. [92] reported that a novel delivery system for curcumin using functionalized single-walled carbon nanotubes by phosphatidylcholine and polyvinylpyrrolidone (SWCNT-Cur) exhibited significantly higher inhibition efficacy on tumor growth and no obvious toxicity in the main organs in a murine S180 tumor model. Moreover, photothermal therapy induced by SWCNTs under near-infrared radiation further facilitated SWCNT-Cur to inhibit the tumor growth in vivo. Wen et al. [93] reported that multi-walled carbon nanotubes (MWNTs) co-delivering sorafenib (Sor) and epidermal growth factor receptor (EGFR) siRNA (MWNT/Sor/siRNA) were proved to show increased Sor release, high siRNA stability and enhanced cellular uptake in liver cancer (LC). However, CNTs exert various toxic effects on the body due to their unique physical and chemical properties. CNT-induced toxicity is considered to be related to surface modification, degree of aggregation in vivo and nanoparticle concentration [94].

(4)Metal nanoparticles

Metal nanomaterials have several therapeutic advantages over other nanomaterials. Firstly, the synthesis of metal nanomaterials allows investigators to change the size, morphology and surface physicochemistry. Secondly, metal nanomaterials can be used as therapeutic agents or imaging agents, or both, so they can easily be functionalized for more complex treatments. Thirdly, a large number of metals can be integrated into heterogeneous structures to show synergistic effects [95]. AuNPs have become the most widely studied NPs due to their excellent properties. Due to space constraints, this section only discusses AuNPs.

The potential of AuNPs as a radiosensitizer has also been investigated. AuNPs are generally injected into tumor cells, and when an external X-ray source acts upon them, they produce free radicals that harm tumor cells and promote cell death. Molinari et al. [96] demonstrated that AuNPs had a basal radiosensitization ability and that AuNPs conjugated with the pyrazolo [3,4-d]pyrimidine derivative SI306, a c-Src inhibitor, when used in combination with radiotherapy, were more effective in inhibiting tumor cell growth with respect to AuNPs and free SI306 in the glioblastoma model. Chen et al. [97] employed bovine serum albumin-capped AuNPs as radiation sensitizers and achieved high efficiency for tumor radiation therapy. No obvious toxicity was observed in in vitro and in vivo experiments. Furthermore, the main advantage of using AuNPs as a heat agent over traditional hyperthermia is that the heating only affects the area near the AuNPs and does not damage other healthy tissues or cells. The temperatures used may be significantly higher for a short period of time [98]. As such, AuNP-mediated hyperthermia acted upon target tissues. In addition, Li et al. [99] aimed at developing antibody-functionalized AuNPs to selectively target cancer cells, and reported that AuNPs conjugated with cetuximab (Ctxb-AuNPs) specifically bound to and accumulated in EGFR-overexpressing A431 cells, compared with EGFR-negative MDA-MB-453 cells, and enhanced the effect of proton irradiation in A431 cells but not in MDA-MB-453 cells.

(5)Magnetic nanoparticles

As the magnetism of magnetic NPs can be affected by internal and external factors, any change in the magnetism will affect the delivery and the release of drugs. As such, the design of magnetic NPs responsive to various factors has gained much attention. Bhattacharya et al. [100] designed a dual temperature and pH-responsive polymer that integrated magnetic nanohybrids comprising smart block co-polymers and mixed ferrite NPs, and these magnetic NPs exhibited enhanced release of Doxil at a higher temperature (37 °C) and lower pH (acidic pH 5.0), while retaining their structure under physiological conditions. Parsian et al. [101] studied the release of the anti-tumor drug gemcitabine on chitosan-coated iron oxide NPs. They found that the release rate of gemcitabine at an acidic pH was eight times higher than that at a neutral pH, which was required for administration at the tumor site. Furthermore, the nanodrug system was more virulent than free gemcitabine on a breast cancer cell line (MCF-7).

### 3.2. Application of Nanoplatforms in COVID-19 Treatment

Coronavirus disease 2019 (COVID-19), caused by severe acute respiratory syndrome coronavirus 2 (SARS-CoV-2), has affected human life and health for more than a year. SARS-CoV-2 is a spherical virus, consisting of a single plus-stranded RNA (+ssRNA) genome wrapped in an envelope with spikes [102]. The virus is highly contagious, transmitted mainly through respiratory droplets when sneezing, coughing or talking [103]. SARS-CoV-2 attaches to the angiotensin-converting enzyme 2 (ACE2) receptor of host epithelial cells through the spike glycoprotein (S protein) [104]. Most ACE2 is present in alveolar epithelial type II cells, but it is also expressed on the surface of cells in other organs [105]. SARS-CoV-2 causes “flu-like” symptoms, such as fever, diarrhea, fatigue and sore throat, and in severe cases, acute respiratory distress and even death [106,107]. At the time of writing (11 September 2021), 224,400,492 cases of COVID-19 have been reported worldwide, including 4,617,660 deaths, and the number of infections and deaths is expected to increase further.

As such, there is an urgent need to design effective vaccines to prevent viral infection, as well as anti-viral and protective therapies, to reduce and eliminate the damage caused by this virus. Nanotechnology has been reported to be efficacious against human immunodeficiency virus (HIV), herpes simplex virus (HSV), human papillomavirus (HPV) and other respiratory viruses. Thus, there is no doubt about the potential of nanotechnology [108]. Any link in the whole process of SARS-CoV-2 infecting target cells and causing disease can be targeted by nanodrugs [109].

#### 3.2.1. Prevention of Virus Attachment and Entry Target Cells

Kim et al. [110] confirmed that porous gold NPs successfully inhibited viral membrane fusion by blocking the influenza A virus (IAV) entry process through the conformational deformation of hemagglutinin. As the cell surface receptor heparan sulfate proteoglycan (HSPG) is the region where coronaviruses bind to host cells, Cagno et al. [111] designed anti-viral NPs with long and flexible linkers mimicking HSPG, thereby allowing for effective viral association with VAL repeating units and generating forces that eventually lead to irreversible viral deformation. In addition, because SARS-CoV-2 enters host cells by interacting with ACE2 through the receptor-binding domain (RBD) of the S protein, any drug that disrupts the binding between RBD and ACE2 has the potential to inhibit SARS-CoV-2. For example, Abo-zeid et al. [112] reported that Fe_2_O_3_ NPs formed a more stable complex with the RBD and interacted efficiently with the RBD.

Non-metal nanomaterials have also been considered as potential treatments for COVID-19. Silica nanoparticles (SiNPs) prevent virus particles from infecting host cells by acting as scavengers. In addition, SiNPs are non-toxic when resuspended in aqueous solutions, so they are biodegradable. Lee et al. [113] reported that a glycosaminoglycan mimetic, when attached to the surface of MSNPs, can block HSV attachment and penetration into susceptible cells. The anti-viral activity of selenium nanoparticles (SeNPs) with low toxicity and excellent activity has attracted increasing attention. Li et al. [114] used amantadine (AM) to modify the surface of SeNPs and reported that AM-SeNPs had less toxicity, and they inhibited the ability of H1N1 influenza to infect host cells by suppressing neuraminidase activity. Carbon dots (CDs) have also been reported to inhibit viral invasion in host cells. Ting et al. [115] reported that the cationic CDs could change the structure of a surface protein in porcine epidemic diarrhea virus (PEDV), thereby inhibiting viral entry. Loczechin et al. [116] found that a functional group in carbon quantum dots (CQDs) derived from 4-aminophenylboronic acid interacts with human coronavirus (HCoV)-229E S protein, thereby inhibiting the entry of HCoV-229E.

#### 3.2.2. Prevention of Virus Replication and Proliferation

The inhibition of virus replication is also an important anti-viral strategy. Erasmus et al. [117] have combined a highly stable nanostructured lipid carrier (NLC) with a replicating viral RNA (rvRNA) encoding Zika virus (ZIKV) antigens and demonstrated a single dose as low as 10 ng can completely protect mice against a lethal ZIKV challenge. Jamali et al. [118] reported that chitosan/siRNA NPs were efficiently taken up by Vero cells, leading to inhibition of influenza virus replication, and nasal delivery of siRNA by chitosan-NPs has anti-viral effects and significantly protected BALB/c mice from a lethal influenza challenge. Nie et al. [119] designed a nanoparticle-based inhibitor that has a matched nanotopology to IAV virions and shows heteromultivalent inhibitory effects on hemagglutinin and neuraminidase. The synthesized nanoinhibitor can not only neutralize the viral particle extracellularly and block its attachment and entry to the host cells, but also significantly reduce the virus replication by six orders of magnitude. In addition, nitric oxide (NO) is an important signaling molecule, which has been shown to have an inhibitory effect on infections caused by specific viruses. Akerstrom et al. [120] found that an organic NO donor, S-nitroso-N-acetylpenicillamine, significantly inhibited the replication cycle of SARS-CoV in a concentration-dependent manner and that NO inhibited viral protein and RNA synthesis.

#### 3.2.3. Inactivation and Elimination of Viruses

Metal NPs can be used as therapeutic drugs due to their local surface plasmon resonance effect [121]. The effects of metal NPs as virus inactivators have been demonstrated in many viruses, including H1N1 and H5N1 influenza viruses [122]. In addition, Wang et al. [123] reported that an artificial nanozyme used in the treatment of hepatitis C virus (HCV) infections could mimic the function of the RNA-induced silencing complex (RISC) machinery for inducing target RNA cleavage. Kong et al. [124] reported that nanodisks carrying the viral receptor sialic acid can rupture the viral envelope irreversibly, trapping viral RNAs inside the endolysosome for enzymatic decomposition. These anti-virals may be applicable to other enveloped viruses such as SARS-CoV-2.

#### 3.2.4. Delivery of Anti-Viral Drugs

Some “re-purposed” drugs, such as remdesivir, chlorohydroquinone, ribavirin and arbidol, are being used to treat COVID-19 patients. However, due to the serious side effects of these drugs, their application is limited [125,126].

Due to the sustained and controlled release of nanomaterials, the pharmacokinetic parameters of the encapsulated drug can be altered, leading to an increased exposure time and a longer therapeutic response, thereby reducing the required drug concentration. Kondel et al. [127] demonstrated the controlled release of solid-lipid NPs coated with acyclovir in a mouse model infected with HSV-1.

NPs eliminate the premature degradation of their loaded drugs in the biological environment. Polyethylene glycol (PEG), PEG-like and other hydrophilic polymers with long chains have a flexible nature that helps to reduce opsonin adsorption on NPs, rendering them unrecognizable by the elimination mechanisms in our body [128]. In addition, small interfering RNAs (siRNAs) that target specific viral genes are a promising therapeutic alternative in the treatment of virus infections. However, siRNAs are vulnerable to degradation by serum nucleases and rapid renal excretion due to their small size and anionic character. Paul et al. [129] reported that cationic AuNP–siRNA complexes significantly reduced dengue virus replication and virion release.

Nanocarriers can deliver high concentrations of anti-viral drugs to target organs and tissues and reduce the inflammatory response. To eliminate the adverse effects of anti-viral drugs in COVID-19 patients, target-specific delivery to the primary target organs and tissues in COVID-19 patients, as well as goblet cells and ciliated cells, is fundamental [130]. Chen et al. [131] reported that a trimethyl chitosan (TMC)-CSKSSDYQC (CSK) peptide conjugate could enhance the oral bioavailability of gemcitabine due to its ability to target intestinal goblet cells and promote intestinal cellular uptake.

Meanwhile, direct pulmonary administration allows rapid absorption due to the high vascularization and circumvention of first-pass metabolism [132]. Thus, inhaled pulmonary delivery may be an ideal route for the delivery of drugs to infected lung epithelial cells [133]. Airborne chitosan NPs can penetrate alveolar epithelial type II cells (AECII) that reside deep within the lungs, ensuring controlled release and minimizing toxicity [134]. Chitosan NPs coat several drugs in the form of an aerosol that sticks to epithelial cells, thereby releasing drugs for up to three hours [135]. Therefore, the anti-viral drug delivery system based on NPs is a non-invasive, practical, efficient and safe means of drug delivery.

#### 3.2.5. Cytokine Storm Prevention

When SARS-CoV-2 is released from infected cells, it offloads the damage-related molecular pattern (DAMP), which signals nearby epithelial cells and alveolar macrophages to secrete pro-inflammatory factors. In most cases, this process triggers an immune response that eliminates the viruses before they spread further. However, in a dysfunctional immune response, multiple inflammatory mediators and chemokines are released, leading to cytokine storms [136].

The first strategy is to prevent abnormally secreted cytokines from contacting target cells, thereby blocking the downstream inflammatory response. For example, Leuschner et al. [137] designed an optimized lipid NP and a CCR2-silencing siRNA that showed rapid blood clearance, accumulation in the spleen and bone marrow and localization in monocytes. Interestingly, the efficient degradation of CCR2 mRNA in monocytes prevents their accumulation at sites of inflammation.

Another strategy is to quickly remove pro-inflammatory cytokines from the blood before they cause further adverse effects on the tissue. Presser et al. [138] reported that hierarchical carbon materials with tuned porosity can effectively adsorb cytokines up-regulated by cytokine storms, such as TNF-α and IL-6, and prevent death due to an uncontrolled inflammatory cascade. Zheng et al. [139] explored the use of graphene nanoplatelets for the rapid removal of a broad spectrum of pro-inflammatory cytokines. The material was less cytotoxic and showed faster adsorption compared with other carbon-based nanomaterials.

NPs can also be used to deliver anti-inflammatory drugs to inflammatory sites, thereby inhibiting cytokine storms. As TNF-α is considered to be one of the major cytokines involved in cytokine storms, the inhibition of TNF-α reduced tissue damage caused by cytokine storms [140]. Aldayel et al. [141] reported that an acid-sensitive and sheddable PEGylated solid-lipid NP formulation of TNF-α-siRNA had a high siRNA encapsulation efficiency and a minimum burst release of siRNA, and it increased the delivery of the siRNA to sites of chronic inflammation in a mouse model. In addition, adenosine is also widely used in the treatment of cytokine storms. However, the direct injection of adenosine causes adverse reactions and serious side effects [142]. Dormont et al. [143] reported that multi-drug NPs prepared by conjugating squalene to adenosine and then encapsulating a-tocopherol, delivered the therapeutic agents in a targeted manner by exploiting endothelial dysfunction at sites of acute inflammation, reducing levels of pro-inflammatory cytokines and increasing levels of IL-10 in mice with acute hyperinflammation and cytokine storm.

Several nanomaterials have been explored for their specific effects on different immune cell subsets. Pentecost et al. [144] demonstrated that nanodiamond can serve as an effective innate immune regulatory platform and that a low-dose ofoctadecylamine-functionalized nanodiamond (ND-ODA) and dexamethasone (Dex)-adsorbed ND-ODA reduced macrophage infiltration and the expression of the pro-inflammatory mediators iNOS and TNF-α.

#### 3.2.6. Vaccine Development

As SARS-CoV-2 is too infectious to be controlled by anti-viral therapy alone, vaccines that harness the innate ability of the immune system to form a protective long-term immune memory to prevent and control the spread of the virus need to be developed [145] (Figure 4).

The antigenic portion of the virus can be encapsulated within a nanocarrier or be coupled to the surface of an NP and applied to the target with adjuvants [146]. As NPs can also serve as adjuvants, the individuals vaccinated with the nanovaccine showed a faster and longer-lasting immune response [147]. In addition, targeted modifications of the NP–antigen conjugates can further enhance the efficacy of the vaccine.

NPs can also be used as a delivery agent for mRNA and DNA vaccines as a means of protecting them from enzymatic degradation and increasing the uptake of genetic material by target cells, thereby overcoming bottlenecks in their use in vivo [148].

Lastly, NPs can induce a cellular and humoral immune response and play an immunomodulatory role. Ma et al. [149] reported that amino-graphene oxide and conventional adjuvant alum could activate the NF-κB pathways, thereby promoting M1 polarization which is associated with the enhanced production of inflammatory cytokines and the recruitment of immune cells. Thomas et al. [150] demonstrated that nucleophile-containing NP surfaces activated complement and became functionalized in situ with C3 upon serum exposure via an alternative pathway. In addition, virus-like particles (VLPs) have been shown to fight viruses effectively by improving the efficiency of APCs. These results were found in studies of both MERS-CoV and SARS-CoV, so it is hopeful that VLPs can be used to effectively treat SARS-CoV-2 infection [151].

### 3.3. Application of Nanoplatforms in Cardiovascular Disease Treatment

Cardiovascular diseases (CVDs) are pathophysiological conditions of the cardiovascular system, and they include atherosclerosis, cardiomyopathy, arrhythmias, myocardial infarction, coronary heart disease, aneurysm and hypertension [152]. CVDs have become a serious public health problem, with the highest morbidity and mortality rates worldwide [153]. As such, it is important that new drugs be developed to treat CVDs. Due to recent advancements in nanoscience and the excellent performance of nanomaterials, nanotechnology has become a new way to treat CVDs [154].

#### 3.3.1. Nanotherapy of Atherosclerosis

(1)Background

Healthy arteries are elastic, but their walls can harden over time. The lesions of the affected arteries start from the intima, with local lipid accumulation, fibrous hyperplasia and calcinosis, forming plaques in a process known as atherosclerosis due to the yellow appearance of the accumulated lipids in the arteries [155]. It is the most common disease of cardiovascular and cerebrovascular systems, mainly affecting large and medium arteries in the body, such as coronary arteries, carotid arteries, cerebral arteries and renal arteries [156]. Furthermore, atherosclerosis is a multi-etiological disease, in that it is caused by multiple factors at different stages of life [157].

(2)Nanoplatforms for treating atherosclerosis

Angiogenesis is an important process in the growth and progression of atherosclerosis. Several anti-angiogenic factors and drugs can be used in the treatment of atherosclerosis. For example, αvβ3 integrin, a biomarker of angiogenesis, is expressed when the endodermis remains physiologically intact. To achieve higher specificity, anti-angiogenic drug perfluorocarbons were designed as phospholipid bilayer nanoemulsions that targeted αvβ3 integrin [158].

As CCR2 is highly expressed in atherosclerotic plaques, Mog et al. [159] reported that self-assembled, peptide-conjugated NPs targeted CCR2^hi^Ly6C^hi^ inflammatory monocytes in the blood and atherosclerotic plaques, resulting in the cell-specific transcriptional down-regulation of key inflammatory genes.

Native low-density lipoproteins (LDLs), which accumulate at atherosclerotic lesions, are the main drivers of atherosclerosis progression. Brusini et al. [160] reported that squalene NPs could accumulate in the aorta of atherosclerotic animals. As such, squalene bioconjugation can serve as an efficient targeting platform for atherosclerosis by accumulating LDL in endogenous NPs. In addition, because of the excellent anti-atherosclerotic properties of high-density lipoproteins (HDLs), much attention has been paid to the design of HDL-mimicking NPs to detect, target and treat atherosclerosis [161].

The high SPR effect of AuNPs is useful in photothermal therapy for atherosclerosis. AuNPs promote macrophage depletion by converting light energy into heat energy in atherosclerosis, thereby decreasing the risk of cardiovascular death by reducing total atherosclerotic volume. AuNPs are also promising carriers of photosensitizers in photodynamic therapy (PDT). AuNPs facilitate the binding of photosensitizers to atherosclerotic plaques, mediating plaque depletion by producing reactive oxygen species during irradiation [162]. As such, the surface modification and functionalization of AuNPs further improved the therapeutic effect. Han et al. [163] investigated the effect of PDT, which was mediated by the up-conversion of fluorescent nanoparticles that encapsulated chlorin e6 (UCNPs-Ce6), on the cholesterol efflux of THP-1 macrophage-derived foam cells, and reported that PDT significantly enhanced the cholesterol efflux and the induction of autophagy in both THP-1 and peritoneal macrophage-derived foam cells, indicating an antagonistic effect on atherosclerosis.

Liposomes are also being actively developed for the treatment of atherosclerosis. Shiozaki et al. [164] reported that cholesterol-rich nanoemulsion (LDE)-paclitaxel was tolerated by patients with cardiovascular disease and reduced the atherosclerotic lesion size. Oumzil et al. [165] demonstrated that the use of nucleoside lipids could form stable solid-lipid NPs loaded with prostacycline, which could inhibit platelet aggregation.

#### 3.3.2. Nanotherapy for Aneurysms

(1)Background

Aneurysms are defined as the localized dilation of an artery that is more than 50 percent of its normal diameter, and they occur mainly in the elderly population [166]. Congenital structural abnormalities or acquired pathological changes of the arterial wall cause local weakness and hypotonia of the vascular wall, and under the continuous impact of blood flow, permanent abnormal dilation or bulging forms [167]. Some factors can accelerate the formation of aneurysms, such as hypertension, which can increase the pressure on the arterial wall, as well as some endocrine factors during pregnancy, which can cause retrogression in the artery wall [168].

(2)Nanoplatform for aneurysm treatment

In some cases, a stent needs to be placed at the site of the aneurysm to isolate it from the blood flow, thereby normalizing the blood flow and slowing the progression of the disease. Wu et al. [169] prepared metallic stents covered with heparin-loaded poly(L-lactide-co-caprolactone) nanofibers, and showed that the nanofibrous matrix would not rupture with the expansion of the metallic stent, effectively separating the aneurysm dome from the bloodstream. Li et al. [170] provided preliminary proof for the effectiveness and the feasibility of nanoelectro-spinning covered stents for the treatment of intracranial aneurysms.

Aneurysms can up-regulate the expression of matrix metalloproteinases, which leads to the degradation of elastin and the elastin matrix in the aortic wall and the progressive loss of elasticity [171]. Therefore, the regulation of matrix metalloproteinases is a feasible treatment approach to delay the growth and progression of aneurysms. Sivaraman et al. [172] demonstrated that surface functionalization of Doxil-loaded poly(lactic-co-glycolic acid) NPs with cationic amphiphiles up-regulated the activity of the elastin crosslinking enzyme lysyl oxidase. In addition to the Doxil released from the NPs involved in the inhibition of MMP-2 production and activity, and the surface functionalization of NPs, cationic amphiphiles may also play a role in MMP-2 inhibition. Yoshimura et al. [173] designed a drug delivery system that could reduce the expression of MMP-9 in the mouse aorta.

#### 3.3.3. Nanotherapy for Myocardial Infarction

(1)Background

Myocardial infarction is an acute condition in which the heart is damaged due to the lack of oxygen-carrying blood caused by acute blockage of the coronary artery [174]. Therefore, the basic cause of myocardial infarction is that the myocardial cells are still working, triggering an imbalance between oxygen supply and demand, leading to necrosis of the myocardial cells [175]. Any factor that may induce thrombosis, plaque shedding, coronary artery spasm or stenosis has the potential to cause myocardial infarction [176].

(2)Nanoplatform for myocardial infarction treatment

Metal and metal oxide-based inorganic NPs have attracted much attention in the treatment of myocardial infarction. Ahmed et al. [177] demonstrated that AuNPs significantly improved isoproterenone-induced myocardial infarction by maintaining myocardial cell structure and the expression of endothelial nitric oxide (eNO) and B-cell lymphoma-2 (Bcl-2). Zheng et al. [178] reported that MnO-based dual-modal NPs preferentially accumulated in the myocardial infarction, which makes them an ideal drug vehicle for the treatment of this condition.

The efficient delivery of antioxidants into mitochondria of ischemic cardiomyocytes where reactive oxygen species are produced is a major challenge for the treatment of myocardial ischemia–reperfusion injury. Cheng et al. [179] designed smart dual-shell polymeric NPs, MCTD-NPs, which deliver the reactive oxygen species scavenger resvertrol specifically to mitochondria of ischemic cardiomyocytes (Figure 5). In addition, insulin-like growth factor (IGF-1) regulates cardiomyocyte function, growth and survival [180]. Chang et al. [181] reported that intramyocardial injection of IGF-1-complexed PLGA NPs increased IGF-1 retention, induced Akt phosphorylation and provided early cardioprotection after acute myocardial infarction.

Thrombus formation is involved in the pathophysiology of myocardial infarction. Absar et al. [182] reported that tissue plasminogen activator (tPA)-loaded liposomes, which were decorated with a peptide sequence of the fibrinogen γ-chain that binds to GPIIb/IIIa expressed by activated platelets, caused a 35% increase in clot lysis and produced a 4.3-fold decrease in the depletion of circulating fibrinogen compared to native tPA, thereby showing favorable physical characteristics and colloidal stability.

An inhibition of extracellular matrix (ECM) degradation is a novel approach in the protection of the heart from ischemia. Cuadrado et al. [183] targeted extracellular matrix metalloproteinase inducer (EMMPRIN) with paramagnetic/fluorescent micellar NPs conjugated with the EMMPRIN-binding peptide AP-9 to inhibit metalloproteinase protein expression and treat myocardial infarction.

Myocardial infarction induced by cardiac ischemia–reperfusion injury is associated with increased intracellular reactive oxygen species and Ca^2+^ levels following reperfusion injury. Hardy et al. [184] demonstrated that multi-functional poly(glycidyl methacrylate) (PGMA) NPs containing curcumin and the AID peptide effectively decreased oxidative stress and superoxide production in cardiac myocytes to modulated reactive oxygen species and Ca^2+^ levels.

## 4. Conclusions and Future Perspectives

Due to rapid advancements in nanotechnology, functionalized nanoplatforms are being widely used in the diagnosis and treatment of various diseases. Specific targeting of two of the most serious diseases, cardiovascular disease and cancers, as well as COVID-19, which is now devastating the global healthcare system, using functionalized nanoplatforms present unique advantages that can overcome bottlenecks in traditional strategies.

The functionalized nanoplatform is a burgeoning field in medical research. Most advancements in this field have focused on diagnosing and treating diseases, but it is also expanding into other areas of medicine, such as antibiotic resistance and artificial organ transplantation. Although functionalized nanoplatforms have shown great potential, most research is still in the early stages, and there are still many unanswered questions. For example, the possible toxicity of nanomaterials and the metabolic problems of NPs require further studies. In addition, the further development of this field will face significant challenges at the regulatory level if related guidance remains unclear and unconsolidated [185]. Therefore, we must not dwell on what has been achieved to date; instead, we must think outside of the box. However, despite the enormous challenges, we believe that functional nanoplatforms will become powerful tools in the fight against various diseases in the near future.

## Figures and Tables

**Figure 1 nanomaterials-11-03346-f001:**
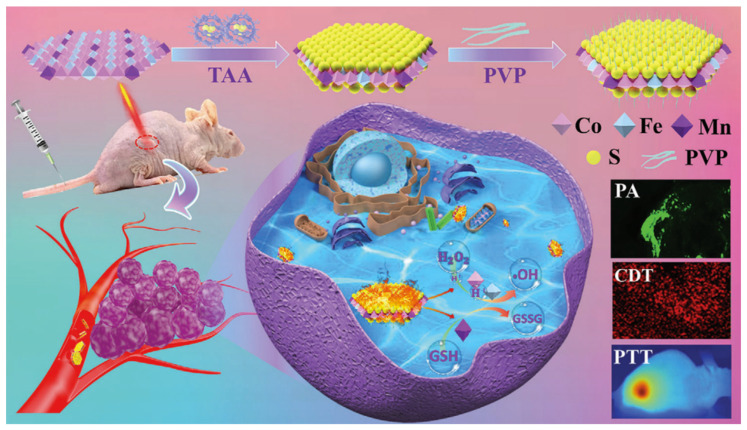
A schematic illustration for preparing CoFeMn dichalcogenide nanosheets to give efficient photothermal therapy and photoacoustic imaging [12].

**Figure 2 nanomaterials-11-03346-f002:**
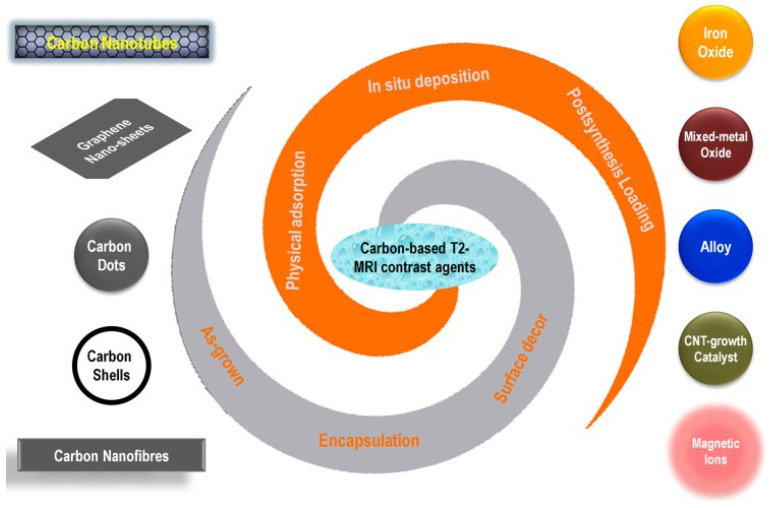
Schematic illustration of various carbon nanoallotropes, magnetic species and their conjugation strategies used for the preparation of T2-weighted MRI contrast agents [27].

**Figure 3 nanomaterials-11-03346-f003:**
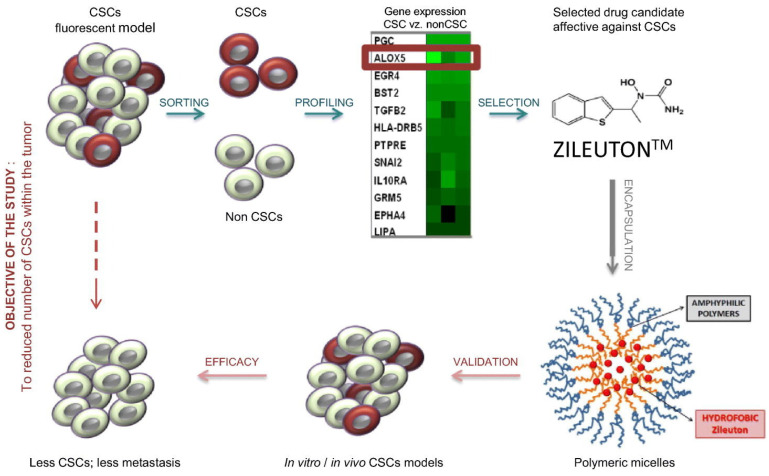
Gener et al. developed and synthesized polymeric micelles loaded with Zileuton™, a potent inhibitor of ALOX5 in cancer stem cells (CSCs), which were chosen as a therapeutic target candidate based on high-throughput screening data. Its great potential for CSC treatment was subsequently demonstrated in vitro and in vivo in breast CSC fluorescent models. PM-Zileuton™ shows strong reduction in the amount of intratumoral CSCs and, further, of blood circulating CSCs in vivo [82].

**Figure 4 nanomaterials-11-03346-f004:**
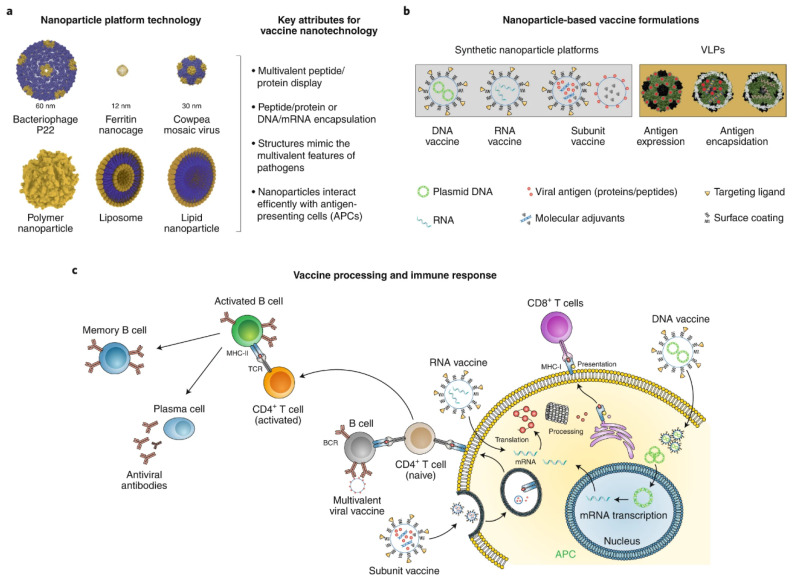
Nanoparticle platform vaccine technologies [145].

**Figure 5 nanomaterials-11-03346-f005:**
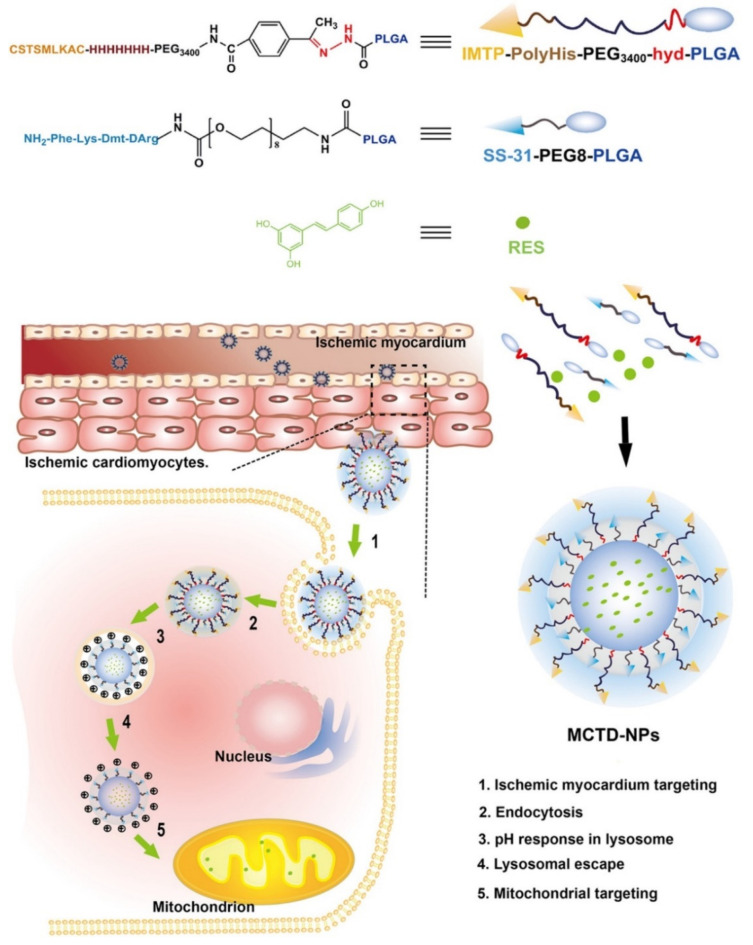
Schematic image of precise treatment of myocardial ischemia–reperfusion injury through multi-stage mitochondrial delivery of resvertrol by using MCTD-NPs [179].

## Data Availability

Not applicable.

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
