# Peer review of "Nanomaterials as Promising Theranostic Tools in Nanomedicine and Their Applications in Clinical Disease Diagnosis and Treatment"

_nanomaterials, 2021, doi:10.3390/nano11123346_

Round 1
Reviewer 1 Report
Dear authors,
I read the material and here are my comments:
- Which is the novelty ot this review with respect to existent literature?
- Which is your contribution to the review in terms of experimental data (own research)?
- I would add more images to sustain the data for various applications (for example vaccine development-scheme or something similar).
- I would strenghten the chapter with COVID-19 with more data. This is very important in the pandemic context.
Reviewer 2 Report
Dear authors,
your review article offers a good overview of applications of nanotechnology in medicine (therapy) with a good and current review of main Nanomaterials and nano platforms used in Nanomedicine.
since many similar reviews have published in the field, I think it will be interested to highlight the pertinence of your review by pointing out its originality.
also, since your review must report an updated in the field of Nanomaterials and nano platforms used in medical nanotechnology, I strongly suggest to cite the following references:
Eventually, toxicity aspects of Nanomaterials shall be better highlighted…
1- Batool Amna ,Menaa Farid ,Uzair Bushra ,Khan Ali Barkat ,Menaa Bouzid, “Progress and Prospects in Translating Nanobiotechnology in Medical Theranostics”, Current Nanoscience 2020; 16(5) . https://doi.org/10.2174/15734137156661911260932582- Bukhari, S.I.; Imam, S.S.; Ahmad, M.Z.; Vuddanda, P.R.; Alshehri, S.; Mahdi, W.A.; Ahmad, J. Recent Progress in Lipid Nanoparticles for Cancer Theranostics: Opportunity and Challenges. Pharmaceutics 2021, 13, 840. https://doi.org/10.3390/pharmaceutics13060840
3-Iqbal H, Khan BA, Khan ZU, Razzaq A, Khan NU, Menaa B, Menaa F. Fabrication, physical characterizations and in vitro antibacterial activity of cefadroxil-loaded chitosan/poly(vinyl alcohol) nanofibers against Staphylococcus aureus clinical isolates. Int J Biol Macromol. 2020 Feb 1;144:921-931. doi: 10.1016/j.ijbiomac.2019.09.169. Epub 2019 Nov 5. PMID: 31704336.
5-Đorđević, S., Gonzalez, M.M., Conejos-Sánchez, I. et al. Current hurdles to the translation of nanomedicines from bench to the clinic. Drug Deliv. and Transl. Res. (2021). https://doi.org/10.1007/s13346-021-01024-2
6- Menaa, F.; Abdelghani, A.; Menaa, B. Graphene nanomaterials as biocompatible and conductive scaffolds for stem cells: Impact for tissue engineering and regenerative medicine. J. Tissue Eng. Regen. Med. 2015, 9, 1321–1338.Best,
the reviewer
Reviewer 3 Report
In this review manuscript, Xisheng Weng and co-workers summarize the recent data on nanomaterials for clinical applications. The abstract is written well and covers the general topics of the review.
Unfortunately, the presented manuscript is very challenging to read and follow due to the several observations:
- The chosen title for this review is very unclear, in particular, the “…their applications in clinical disease” seems unfinished and I would suggest the “Nanomaterials as promising theranostic tools in nanomedicine and their clinical applications” or “Nanomaterials as promising theranostic tools in nanomedicine and their applications in clinical disease diagnosis and treatment” as possible titles for the review.
- Analyzing the manuscript, I’ve started with the references list and observed a big number of incomplete references, with very different styles. This makes the checking of the sources almost impossible, making the information in review uncertain. For example, references 5, 7, 8, 10, 11, 12, 14, 15, and so on till the end of the list do not have either the pages, or volumes, or both indicated.
- The introduction section is very small with only general information with no clear topics to be covered in the review. No references on previous reviews in this popular field are mentioned, and what different specific topics are covered by this particular review in comparison to other already published ones. Also, the incorrect construction of some sentences in the introduction makes it very difficult to follow. For example lines 37-39 “Nanomedicine is the continuous development and introduction of new disciplines that are derived from nano diagnostic technologies, nano-drugs, nano-robots, nano-safety evaluation, and other fields”.
- The Gold-based nanoparticle section (lines 68-83) gives absolutely unclear information. One could guess that discussed gold nanoparticle design probably includes aptamers. Very unclear section.
- The introduction to Covid-19 treatment section (lines 420-439) is too detailed, with many references and Figure 3 which have nothing in common with the current review topic.
- Use of acronyms which are not specified in the text, for example, SERS (line 99), SPION (line 120), POCT (line 157), NGS (line 179) – abbreviation explained, but never used again in the text, CSC (line 329), Mesoporous silica NPs (MSNs) in line 352 appears as mesoporous silica NPs (MSNPs) in line 362, SiNPs (line 459), and so on till the end of the text.
The above observations suggest that the manuscript is not properly prepared and needs considerable improvement. Major revisions are suggested prior to the next reviewing process.
Reviewer 4 Report
Wei Zhu et al. have written a well balanced review on an important and rapidly expanding field of the applications of nanomaterials in the diagnosis and treatment of diseases.
Minor objections:
(1) Only the last sentence in Abstract refers to the content of the review, the rest actually belongs to Introduction. Abstract should be re-written in order to make explicit what are the topics covered in the review;
(2) There are numerous published reviews of similar kind and the authors should state the time period they have covered and emphasize the reviews to which their review is a chronological extension. Putting differently, what were the reasons to offer a review such as this one at this particular time?;
(3) "Fe iron-based nanoparticles" should be replaced by "Iron-based nanoparticles".
Round 2
Reviewer 3 Report
Comments to the Authors:
In the revised manuscript, Xisheng Weng and co-workers have succeeded to considerably improving the Abstract and the Introduction sections, together with the title of the presented review. The improvements have been also made to all sections. Still, there is a number of important changes that have to be made:
- Line 149: should be Forster Resonance Energy transfer.
- Reference 12 - Advanced Science 2020 is incomplete.
- Reference 16 is described in detail in the text, but it is from 2014, while the introduction states that the Review focuses on the advances in the field from the last 5 years.
- Line 191-192 is displaced.
- In section 2.3, references 27, 29, 32 33, 34, 36 and 37 are from 2011-2014, so again, their description in the Review is out of date (not in the last 5 years).
- Line 405: CTC is an abbreviation without description.
- In section 3.1.1 the discussed references 57 and 58 are from 2011, out of date.
- (2) Particle shape all references (59-62) are out of date (2007-2014).
- Discussed reference 67 is out of date (2005).
- Reference 81 and 82 are in detail discussed but out of date (2009, 2010).
- Reference 86 in detail discussed but out of date (2011).
- Reference 92 and 94 are in detail discussed but out of date (2010, 2014).
- Reference 97 in detail discussed but out of date (2007).
- The information between lines 523-529 including Figure 4 is not important in the view of the review topic and should be deleted.
- Title of 3.2.1. Prevent virus attachment and entry into target cells seems grammatically incomplete. Prevention of virus attachment and entry target cells could be a substitute.
- Reference 108 in detail discussed but out of date (2010).
- Line 554: CD abbreviation is not indicated.
- Reference 117 in detail discussed but out of date (2005).
- All references (lines 871 – 1374) should possess complete information and be written in the same style. Too many references have no publication year, or volume and pages, authors are missing. This makes the checking of the sources almost impossible, making the information in review uncertain.
Author Response
Dear Reviewer #3,
Thank you very much for your valuable comments on our manuscript, nanomaterials-1421690, entitled "Nanomaterials as promising theranostic tools in nanomedicine and their applications in clinical disease". Your comments make us realize the inadequacy of our work and also bring us very important inspirations. Please allow me to report to you the details of our revision of this manuscript.
- As you pointed out that the information of some references is not complete, we re-checked all the references, downloaded the links of the references from Web of Science and re-entered them into the manuscript through Endnote 20. We have ensured the integrity of the vast majority of reference information in this way. However, we could not find information on the volume or pages of some references. We believe that this information is missing, because of the journals themselves.
- We are very sorry that we did not strictly eliminate outdated references when writing the manuscript. We have complied with your request to remove most of the outdated references and to add up-to-date references accordingly to ensure the timeliness of this manuscript. However, for the content of "Particle shape", we did not find more appropriate references, because there are few researches in this field in recent years. In our opinion, this section does not attempt to introduce readers to recent researches. The significance of this section is mainly to ensure the integrity of the content and the tightness of logic. So, can we cite outdated but appropriate references to illustrate our point? In addition, we have changed the "based on published research in the last five years" in the abstract to "based mainly on published research in the last five years".
- We appreciate your careful review of this manuscript and the discovery of many undetected errors. We respect your professionalism and have corrected the errors you found.
We deeply appreciate your important comments on the revision of our manuscript, and we hope you are satisfied with our revision. We are very grateful for your help!
With kind regards,
Xisheng Weng, MD
Department of Orthopaedics, Peking Union Medical College Hospital, Chinese Academy of Medical Sciences & Peking Union Medical College, Beijing 100730, China
E-mail: xshweng@medmail.com.cn

Round 3
Reviewer 3 Report
Comments to the Authors:
In the revised manuscript, Xisheng Weng and co-workers have succeeded to considerably improve the manuscript by updating the literature and correcting previously mentioned observations.
The Reference list is still full of incomplete references, including their own reference 12. Please, fill in and arrange the references accordingly. The list of incomplete references: 12, 13, 17, 19, 23, 24, 25, 27, 32, 33, 37, 38, 42, 47, 49, 50, 55, 57, 71, 74, 85, 86, 87, 90, 92, 94, 105, 107, 112, 113, 115, 125, 127, 133, 138, 142, 145, 146, 149, 151, 154, 155, 158, 163, 164, 165, 167, 176, 178, 188.
Author Response
Dear Reviewer #3,
Thank you very much for your valuable comments on our manuscript, nanomaterials-1421690, entitled "Nanomaterials as promising theranostic tools in nanomedicine and their applications in clinical disease". Your comments make us realize the inadequacy of our work and also bring us very important inspirations. Please allow me to report to you the details of our revision of this manuscript.
In accordance with your request, we have completed the information of the references, including their page numbers. Some references do not have information about page numbers, so we use Article Number instead of Page Number. However, Article Number instead of Page Number for ref. 158 and 188 was not available.
We deeply appreciate your important comments on the revision of our manuscript, and we hope you are satisfied with our revision. We are very grateful for your help!
With kind regards,
Xisheng Weng, MD
Department of Orthopaedics, Peking Union Medical College Hospital, Chinese Academy of Medical Sciences & Peking Union Medical College, Beijing 100730, China
E-mail: xshweng@medmail.com.cn
